# Affinity Resins for the Isolation of Immunoglobulins G Obtained Using Biocatalytic Technology

**DOI:** 10.3390/ijms25031367

**Published:** 2024-01-23

**Authors:** Mikhail N. Tereshin, Tatiana D. Melikhova, Barbara Z. Eletskaya, Olga B. Ksenofontova, Pavel V. Pantyushenko, Maria Ya. Berzina, Igor Ivanov, Igor V. Myagkikh, Vasiliy N. Stepanenko

**Affiliations:** 1Shemyakin-Ovchinnikov Institute of Bioorganic Chemistry, Russian Academy of Sciences, Miklukho-Maklaya St. 16/10, 117437 Moscow, Russia; misha060596@gmail.com (M.N.T.); tdm-63@yandex.ru (T.D.M.); fraubarusya@gmail.com (B.Z.E.); labpeptos@gmail.com (O.B.K.); pantyushenko.pavel@gmail.com (P.V.P.); myagkikh@ibch.ru (I.V.M.); svn@ibch.ru (V.N.S.); 2Lomonosov Institute of Fine Chemical Technologies, MIREA—Russian Technological University, Vernadskogo pr. 86, 119571 Moscow, Russia; igor_ivanov@gmx.de

**Keywords:** protein A chromatography, monoclonal antibodies, sortase A

## Abstract

Affinity chromatography resins that are obtained by conjugation of matrices with proteins of bacterial origin, like protein A, are frequently used for the purification of numerous therapeutic monoclonal antibodies. This article presents the development of a biocatalytic method for the production of novel affinity resins with an immobilized mutant form of protein A via sortase A mediated reaction. The conditions for activation of the agarose Seplife 6FF matrix, selection of different types of linkers with free amino groups and conditions for immobilization of recombinant protein A on the surface of the activated matrix were studied. Finally, the basic operational properties, like dynamic binding capacity (DBC), temperature dependance of DBC and stability during the cleaning-in-place process of the affinity resin with the Gly-Gly-EDA-Gly-Gly linker, were assessed using recombinant hyperchimeric monoclonal antibodies. The main characteristics show comparable results with the widely used commercial samples.

## 1. Introduction

According to the latest data, the market for monoclonal antibodies (mAbs) that are used as therapeutic agents for the treatment of various diseases like oncology, multiple sclerosis, rheumatoid arthritis or Crohn’s disease is steadily growing [1,2,3]. In the USA and Europe, about 140 monoclonal antibodies either have been approved as therapeutics by 2023 or are in the clinical trial phases 2 and 3. Affinity chromatography that utilizes resins with conjugated protein A from *Staphylococcus aureus* is frequently applied for the isolation of recombinant monoclonal antibodies. This ligand is characterized by high specificity for the Fc fragment of immunoglobulins G [4]. In most of the classical protein A conjugation methods that are used for the preparation of affinity resins, protein ligand immobilization is carried out by the reaction of lysine side-chain amino groups with activated groups on the carrier matrix. Such nonspecific amide bond formation may result in a partial loss of protein activity due to its incorrect orientation on the surface. Site-specific conjugation of protein A can be achieved when the protein has a single accessible reactive cysteine (Cys) at its C-terminus. In this case, the amino acid thiol group undergoes nucleophilic substitution to form stable thioether bonds [5].

In our work, we applied an alternative enzymatic approach which allows for the immobilization of recombinant protein A using the sortase A (SrtA, EC 3.4.22.70)-mediated transpeptidation reaction. Sortase A from *Staphylococcus aureus* allows us to perform site-specific immobilization of both proteins and peptides on solid matrixes by in vitro ligation [6,7]. This enzyme recognizes the specific sequence LPXTG (where X is any other amino acid) and cleaves it between the threonine and glycine residues to form an intermediate enzyme–substrate complex via an ester bond between the threonine of the substrate and the active site cysteine of enzyme. The latter is further subjected to a nucleophilic attack by the amino group of oligoglycine (from 2 to 5 residues), resulting in the formation of a new peptide bond between the threonine and glycine (Figure 1) [8].

Immobilization of proteins or peptides on a matrix via sortase A-mediated reaction requires the presence of polyglycine fragments on its surface. The “shortening” of the length of polyglycine to diglycine does not affect the enzymatic activity of sortase A [9]. Moreover, this enzyme also utilizes primary amines as nucleophiles [10]. Kuropka and colleagues demonstrated sortase-mediated ligation of the hSH3N domain of the adapter protein ADAP (11 kDa) to the triglycine-modified agarose matrix. Under these experimental conditions, 0.7–0.8 nmol of protein was immobilized on 20 μL of agarose [11], and these data indicate the utility of transpeptidation in the preparation of resins with protein ligands.

## 2. Results and Discussion

### 2.1. BsrtA Ligand Design

To increase the stability of recombinant protein A (BsrtA) and to allow for its application for affinity purification purposes, we designed a recombinant protein ligand that contains the following elements (Figure 2) [12]:The design utilizes the natural sequence of protein A (domain B), which has a high affinity for the Fc fragment of immunoglobulins;The leader sequence has been introduced to increase the efficiency of recombinant protein biosynthesis;The recombinant protein contains two B domains of protein A followed by each other in order to increase the dynamic capacity of the resin obtained on its basis;Six-point mutations of asparagine were performed in the sequence of domain B to increase the resistance of the recombinant protein in alkaline conditions during the resin’s cleaning-in-place process;The recognition peptide sequence LPETG was introduced before the polyhistidine tag (HisTag), which is necessary for biocatalytic conjugation of the ligand matrix;The HisTag was introduced for affinity purification of the protein during ligand production.

### 2.2. Preparation of Conjugates

An ultimate requirement of the presence of a primary amino group of glycine for the transpeptidation reaction initiated our efforts to look for various linkers and investigate the influence of both the length and the chemical structure of the spacer on the capacity of affinity resins. Model affinity resin A was obtained using commercial matrix Amino-Sepharose 6FF (Cytiva) which itself contains a primary amino group as a linker. Samples of affinity resins B-F were synthesized using various linkers (Figure 1) attached to the 6% cross-linked Seplife 6FF (Sunresin) agarose beads as a matrix. The first stage of the synthesis was activation of the Seplife 6FF matrix through the reaction with epoxychlorohydrin (ECH) (activated matrix **4**) or diglycidyl ether of butanediol-1,4 (BDDE) (activated matrix **5**) (Figure 1). In both cases, activated matrices **4** and **5** contained about 25–35 μM of epoxy groups per 1 mL of matrix. Quantitative determination of the content of attached epoxy groups was carried out using the titration method described in the Materials and Methods section. The samples of the activated matrices were further modified by different linkers with primary amino groups. To obtain sample **7**, the primary amino group was introduced by the reaction of the activated matrix **4** with ethylenediamine (EDA) **6** (Figure 1).

Since oligoglycine sequences are the natural substrates for sortase A, on the next step, we tried to extend the spacer length using Boc-diglycine **8** and Boc-tetraglycine **9**. Boc-tetraglycine **9** was obtained via the corresponding succinimide ester of compound **8**. The procedure of synthesis of samples **10** and **11** with EDA- Gly-Gly and EDA-Gly-Gly Gly-Gly linkers, respectively, included three synthetic steps. In this case, the Boc deprotection stage required the presence of an aqueous solution of trifluoroacetic acid (pH = 2) which negatively affected the matrix characteristics under these hard experimental conditions. An application of the Gly-Gly-EDA-Gly-Gly linker (**13**) (Figure 1) has been proposed as an alternative method for the conjugation of diglycine with an epoxy-activated matrix. Linker **13** was obtained from Boc-diglycine **8** through the reaction of its succinimide ester with ethylenediamine **6**. Using this linker, samples **14** and **15** were obtained on the basis of activities matrices **4** and **5**, respectively. The number of amino groups attached to the matrix with the linker are given in Table 1.

The immobilization of BsrtA ligand on either Amino-Sepharose 6FF or modified Seplife 6FF agarose matrices **7**, **10**, **11**, **14** and **15** was performed by the biocatalytic procedure using heptamutant sortase A [12]. Ca^2+^-independent heptamutant sortase A was obtained as described previously [13,14]. If the dynamic capacity of resin A when reaching 10% breakthrough (DBC_10_) was rather low, the corresponding value for resin B was almost doubled and constituted 40.6 mg IgG per 1 mL of resin with a residence time of 6 min. The data obtained for the samples of resins A and B suggest the ability of sortase A to recognize primary amino groups of compounds other than oligoglycines [10]. Other resins obtained (resins C–F) demonstrated higher DBC_10_ values when compared to those of resins A and B. An increase in the number of glycines in the linker from two to four did not significantly affect the resin capacity (Table 1). Resin E has demonstrated the highest DBC_10_ value (50 mg IgG per 1 mL of resin) among the resins tested. This result is comparable to the capacity of several commercial agarose-based resins, like MabSelect and MabSelect SuRe, which show under our experimental conditions DBC_10_ values of 48.2 and 56.9 mg of IgG per ml of resin, respectively. Finally, 5.9–6.2 mg of recombinant protein A analogue were ligated on 1 mL of resin E and this amount corresponds to 378–397 nM of the ligand attached to 1 mL of the matrix (see Section 3). The conjugation yield is approximately 30%, calculated based on values of immobilized BsrtA (about 6 mg per 1 mL of the resin), and we initially added BsrtA to the reaction mixture (about 20 mg per 1 mL of the resin).

### 2.3. Characterization of Affinity Resin E

In the next stage, we analyzed operational parameters of resin E. First, we tested the dependence of the DBC_10_ value on the residence time (Figure 3A). At the maximum residence time of 10 min, the capacity of resin E reached 55 mg of IgG per 1 mL of resin under our experimental conditions. Another important characteristic of affinity resins is their resistance to the cleaning-in-place (CIP) process, which allows them to be used repeatedly without any significant loss in capacity. As a rule, the CIP is carried out using a 0.1–0.5 M NaOH solution. After 100 cycles, the DBC_10_ value of resin E did not decrease significantly (Figure 3B), and these data indicate the stability of the resin under repeated use conditions.

Moreover, we noticed that DBC_10_ values are greatly influenced by the temperature factor, and these data are in accordance with some previous reports [15]. Thus, the dynamic capacities of both resin E and commercial resin MabSelect were tested at different temperatures (Figure 3C). Interestingly, an increase in the temperature from 20 °C to 30 °C induced almost a 20% increase in the DBC_10_ value of resin E, and these data are comparable to the difference of 23% observed for commercial resin MabSelect Sure. This feature of agarose-based resins must be taken into account when evaluating DBC_10_.

In the industrial production of mAbs, high flow rates are frequently used during the cleaning and regeneration steps. Therefore, we assessed the pre-column pressure in the chromatography system at different linear flow rates (Figure 3D). As can be seen from Figure 3D, even at a flow rate of 600 cm/h, the pre-column pressure did not exceed 0.12 MPa (1.2 bar).

As shown above, sortase A-mediated ligation allows for successful immobilization of the recombinant analogue of bacterial protein A. The resulting resin exhibits high dynamic capacity and stability during CIP. This has been achieved due to the site-specific reaction, which is an undoubtful advantage when compared to other chemical immobilization methods. Thus, the application of some nonspecific chemical methods frequently results in the formation of resins with dynamic capacity which do not exceed 13 mg/mL [16]. Alternatively, Wang and co-authors demonstrated that the introduction of the glutaraldehyde as an additional linker and a change in the immobilization conditions can alter to some extent the protein orientation on the matrix and increase DBC_10_ values (up to 45 mg of IgG per 1 mL of the resin) in resulting resins [17]. Another method of chemical immobilization which utilizes C-terminal cysteine to force site-specific orientation of the ligand [5] has several disadvantages in comparison to our biocatalytic procedure. Although the DBC_10_ value of the resulted resin was 50 mg of IgG per swollen gram, this method required multistage activation of the matrix (epoxy activation, introduction of amino groups, and maleimide functionalization) and blocking of residual reactive groups after immobilization. Moreover, the authors noted the dimerization of ZZZ-Cys in the solution during storage due to the formation of disulfide bonds between the C-terminal protein cysteine molecules. ZZZ-Cys dimers were observed in trace amounts within a day, reaching 50% dimerization within a week, necessitating the restoration of this bond using mercaptoethanol or dithiothreitol and an additional desalination stage for the obtained protein solutions. Furthermore, the stability of the resulted resin in alkaline treatment was not evaluated in this work. In contrast, our method allows for the synthesis of affinity resins within fewer stages, whereas a peptide bond formed between the ligand and the matrix provides high resistance of the resin to CIP. Moreover, the C-terminal part of the hybrid protein, which can carry various auxiliary sequences (like the His-tag in our case), is cleaved off during the transpeptidation reaction.

## 3. Materials and Methods

### 3.1. General

Affinity resins MabSelect and MabSelect SuRe were purchased from GE Healthcare (Uppsala, Sweden), Seplife 6FF matrix from Sunresin (Xi’an, China), Dowex 50W X8 hydrogen from Serva (Heidelberg, Germany), IMAC SepFast™ resin from BioToolomics (Consett, UK), DEAE Sepharose resin from GE Healthcare (Uppsala, Sweden), Sephadex G-25 was purchased from GE Healthcare (Uppsala, Sweden), and HUMIRA^®^ (adalimumab) from AbbVie (Ludwigshafen, Germany). Tris(hydroxymethyl)aminomethane, urea, imidazol, sodium acetate, sodium bicarbonate, sodium chloride and ethylenediaamine tetraacetate were purchased from Merck (Darmstadt, Germany).

^1^H NMR spectra were recorded with a 700 MHz Bruker MSL spectrometer (Germany) in DMSO-*d*_6_. Chemical shifts are given in ppm, with spin–spin interaction constants in Hz. Analytical HPLC-MS was performed on a Waters Acquity UPLC system (Milford, MA, USA) equipped with the Acquity UPLC BEH C18, 1.7 µm, (2.1 × 50 mm) column. Buffer A: 0.1% formic acid in water; buffer B: 0.1% formic acid in CH_3_CN; linear gradient from 5 to 80% of buffer B in buffer A for 8 min at a flow rate 0.5 mL/min. Detection was performed spectrophotometrically at 220 nm and using either the positive (ES^+^) or negative (ES^−^) mode of electrospray ionization. The mass scan regiment was set up from 50 to 1050 Da. DBC_10_ estimation was carried out using a GE AKTA Pure chromatography system (GE Healthcare Bio-Sciences, Uppsala, Sweden) equipped with UNICORN software 7.3 SP1 (GE Healthcare Bio-Sciences, Uppsala, Sweden). The determination of protein concentration was carried out on a UV/visible spectrophotometer, Ultrospec-1000 (Amersham Pharmacia Biotech, Cambridge, UK).

### 3.2. Chemistry

#### 3.2.1. Epoxy-Activated Agarose Derivatives **4** and **5**

Epoxy activation of the Seplife 6FF matrix was carried out using either epichlorohydrin (**2**) or 1,4-butanediol diglycidyl ether (**3**) in accordance with previously described methods [18,19]. Filtered sepharose Seplife 6FF (10 g) was transferred into a 50 mL plastic tube. A total of 5 mL of either epichlorohydrin (ECH) or 1,4-butanediol diglycidyl ether (BDDE), 5 mL of 1 M NaOH and 10 mL of dimethyl sulfoxide were added to the agarose. The tube was placed on a rotating platform and mixed for 2–14 h at 25 °C. At the end of the reaction, the modified agarose was placed on a Schott filter and washed successively with 25% ethanol in H_2_O (100 mL), H_2_O (100 mL) and 100 mM NaHCO_3_ at pH 8.5 (100 mL).

#### 3.2.2. Determination of the Number of Oxirane Groups Attached to the Matrix

The epoxy-activated agarose **4** or **5** (1 g) was placed on a Schott filter and washed with distilled water (50 mL). A total of 10 mL of 1.3 M Na_2_S_2_O_3_ solution was then added to a sample of the matrix. The mixture was thermostated for 30 min at 30 °C. Then, a few drops of an indicator (phenolphthalein) were added and the mixture was titrated with a solution of 0.01 M HCl until the indicator became discolored (pH = 7). The amount of attached oxirane groups was determined by the amount of titrant solution used [20].

#### 3.2.3. Aminoethyl Agarose (**7**)

Epoxy-agarose **4** was kept in a solution of 200 mM ethylenediamine **6** in 100 mM NaHCO_3_ (pH 8.2) with stirring for 8–12 h at 25 °C. The aminoethyl agarose was placed on a Schott filter and washed successively with H_2_O (100 mL) and 100 mM NaHCO_3_ at pH 8.5 (100 mL).

#### 3.2.4. Boc-Gly-Gly-Gly-Gly-OH (**9**)

Boc-protected tetrapeptide **9** was prepared from Boc-protected diglycine (Sigma-Aldrich, St. Louis, MO, USA) **8** via an intermediate succinimide ester. To a pre-cooled −5 °C solution of Boc-Gly-Gly-OH (**8**) (23.8 g, 0.10 mol) and N-hydroxysuccinimide (13.8 g, 0.12 mol) in a mixture of 160 mL THF and 15 mL DMF, a pre-cooled solution of N,N′-dicyclohexylcarbodiimide (29 g, 0.14 mol) in 40 mL of tetrahydrofuran was added and the resulting mixture was stirred first for 1 h at −5–0 °C and an additional 18 h at room temperature. At the end of the reaction, the precipitate of dicyclohexylurea was filtered off and then washed with tetrahydrofuran. The combined filtrates were evaporated and the product was crystallized from ethyl ether and dried. The yield of the intermediate product (Boc-Gly-Gly-OSu) was 22 g (67%). A solution of the Boc-Gly-Gly-OSu (3.56 g, 0.011 mol) in a dioxane–water mixture (2:1) was added to a solution of diglycine 1.58 g (0.012 mol) in a mixture of saturated solution of NaHCO_3_ in water (2 mL) and 1N NaOH (1 mL). The reaction mixture was stirred for 18 h at room temperature. To remove impurities, the reaction mixture was filtered through Dowex 50W X8 resin Serva (Heidelberg, Germany). The resulting solution was concentrated under reduced pressure. The solid residue was crystallized from isopropanol and air-dried. Yield of Boc-Gly 4-OH was 2.19 g (49%). ^1^H NMR (700 MHz, DMSO-*d_6_*) δ 8.14 (t, *J* = 5.60 Hz, 1H, NH), 8.07 (t, *J* = 5.60 Hz, 1H, NH), 8.02 (t, *J* = 5.60 Hz, 1H, NH), 6.98 (t, *J* = 5.60 Hz, 0.9H, NH-Boc), 3.76 (d, *J* = 5.90 Hz, 2H, CH_2_), 3.74 (d, *J* = 5.50 Hz, 2H, CH_2_), 3.73 (d, *J* = 5.40 Hz, 2H, CH_2_), 3.59 (d, *J* = 5.90 Hz, 2H, CH_2_), and 1.39 (s, 9H, Boc). The ^1^H NMR spectrum is shown in Appendix A. MS *m*/*z* (ES+): calculated for C_13_H_22_N_4_O_7_ [M + H]^+^ 347.2; found [M + H]^+^ 347.3. 

#### 3.2.5. Boc-Gly-Gly-EDA-Gly-Gly-Boc (**12**)

To a solution of Boc-Gly-Gly-OSu (20 g, 0.06 mol) in 140 mL of dioxane and 10 mL of DMF, ethylenediamine **6** (2 mL, 0.03 mol) was added and the mixture was stirred for 6 h at room temperature. The precipitate formed during the reaction was filtered off, washed with diethyl ether (120 mL) and dried in air. Yield of **12** was 10.8 g (74%). ^1^H NMR (700 MHz, DMSO-*d_6_*) δ 7.98 (t, *J* = 5.76 Hz, 2 H, NH), 7.82 (m, 2 H, NH-C-C-NH), 6.98 (t, *J* = 6.12 Hz, 1.7 H, NHBoc-trans), 6.54 (s, 0.3 H, NH-Boc-cis), 3.67 (d, *J* = 5.71 Hz, 4 H, glycine CH_2_-N), 3.58 (d, *J* = 5.99 Hz, 4 H, glycine CH_2_-N-Boc), 3.11 (m, 4 H, N-CH_2_-CH_2_-N), and 1.39 (s, 18 H, Boc). The ^1^H NMR spectrum is shown in Appendix A. MS *m*/*z* (ES^+^): calculated for C_20_H_36_N_6_O_8_ [M + H]^+^ 489.3; found [M + H]^+^ 489.3.

#### 3.2.6. Gly-Gly-EDA-Gly-Gly (**13**)

A total of 50 mL of 80% trifluoroacetic acid pre-cooled to 0 °C was added to compound **12** (10 g, 0.02 mol) and the mixture was stirred for 1 h at room temperature. After trifluoroacetic acid was evaporated, 20 mL of isopropanol was added to the residue and the solvent was again evaporated to dryness. The resulting solid was dissolved in water, filtered and lyophilized. The product was obtained in the form of a trifluoroacetate in a yield of 9.7 g (97%). MS *m*/*z* (ES^+^): calculated for C_10_H_20_N_6_O_4_ [M + H]^+^ 289.2; found 289.3. MS *m*/*z* (ES^−^): calculated C_12_H_21_N_6_O_6_F_3_ for [M + TFA − H]^−^ 401.1; found 401.2.

#### 3.2.7. General Method for Preparation of polyGly-Seplife 6FF Agaroses **10** and **11**

The amino-agarose matrix **7** (10 g) was washed sequentially with water (5 × 20 mL), a 20% solution of ethanol in water (1 × 30 mL), a 40% solution of ethanol in water (1 × 30 mL), a 60% solution of ethanol in water (1 × 30 mL), and 100% ethanol (3 × 20 mL) and additionally with dimethylformamide. A solution of polyglycine ligand (4 equivalents relative to the number of amino groups), HBTU (4 equiv) and ethyldiisopropylamine (8 equiv) in dimethylformamide were added to the washed matrix **7** (1 equivalent relative to the number of amino groups). The reaction mixture was stirred for 2 h at room temperature. The reaction progress was analyzed using the Kaiser qualitative test [21] to determine the residual free amino groups present on the matrix. After the polyglycine ligand addition reaction had completed, the resulting matrix was washed on a Schott filter using dimethylformamide. To remove the Boc-protecting group, the matrix with the attached Boc-polyglycine was washed with 150 mL of distilled water, and then, 15 mL of 95% trifluoroacetic acid was added and the reaction mixture was stirred for 2 h at room temperature. To remove trifluoroacetic acid, the matrix was repeatedly washed on a Schott filter with water until the pH of the eluate was neutral. The deprotection efficiency was monitored using the Kaiser method.

#### 3.2.8. General Method for Preparation of Gly-Gly-EDA-Gly-Gly-Seplife 6FF-Agarose **14** and **15**

Gly-Gly-EDA-Gly-Gly **13** (0.88 g, 0.030 mmol) was dissolved in 7 mL of 0.1 M NaHCO_3_ and the pH of the solution was adjusted to 9.5 using 1 N NaOH. The resulting solution was added to epoxy-agaroses **4** or **5** (6 g) and the reaction tube was placed on a rotating platform for 24 h at 37 °C. The resulting matrix was washed with a large amount of distilled water (200 mL) and then with 0.1 M NaHCO_3_ (200 mL).

#### 3.2.9. Determination of the Number of Amino Groups on the Matrix

Estimation of the number of amino groups was performed by the titration method. For this purpose, 3 mL of the matrix was placed on a Schott filter and washed first with a solution of 1 M HCl (50 mL) and then with distilled water until the pH of the eluate became neutral. Bound chloride ions were eluted from the sorbent using a 0.6 M sodium nitrate solution. The chloride ion concentration in the eluate was determined by titration with an aqueous solution of AgNO_3_ (0.1 M) in the presence of potassium chromate as an indicator. After the precipitation of silver chloride at the equivalence point, silver chromate was formed, and the yellow color of the solution turned into orange–yellow.

### 3.3. Biotechnology

Preparation of recombinant BsrtA protein. Recombinant domain B of protein A (BsrtA) was prepared as described previously [12]. Briefly, the method involves the preparation of a recombinant protein expression plasmid, selection of an *Escherichia coli* strain capable of producing the BsrtA and protein isolation. After induction, bacterial cells (100 g) were separated by centrifugation (5000× *g*, 20 min, 4 °C) and lysed using an ultrasonic disintegrator (Elma, Schmidbauer GmbH, Singen, Germany) in 700 mL of 50 mM Tris-HCl and 5 mM EDTA at pH 8. After that, urea was added to the lysate to reach the final concentration of 4 M, followed by a consequent dilution of the sample with 50 mM Tris-HCl buffer (pH 8), to reach a final urea concentration of 1 M. The protein was purified on an XK 26/20 column (Cytiva, Life-Sciences, Marlborough, MA, USA) packed with DEAE Sepharose ion exchange sorbent under gradient elution conditions from 0 to 0.5 M NaCl in 50 mM Tris-HCl buffer(pH 8). In the next stage, the IMAC metal chelate sorbent was used XK 26/20 column (Cytiva, Life-Sciences, USA). The protein was eluted with 0.5 M imidazole in 50 mM Tris and 0.3 M NaCl (pH 8). In the final stage, the BsrtA solution was desalted on a Sephadex G-25 XK 50/100 column (Cytiva, Life-Sciences, USA) using 100 mM NaHCO_3_ (pH 8.5) as a buffer.

#### 3.3.1. Preparation of Recombinant Sortase A

Ca^2+^-independent sortase A (heptomutant) was prepared according to the method described previously [13,14]. Briefly, after the induction of sortase A expression, the bacterial pellet (100 g) was separated by centrifugation (5000× *g*, 20 min, 4 °C) and lysed using an ultrasonic disintegrator (Elma) in 50 mM Tris and 0.5 M NaCl (pH 7.5) buffer that contained 10% glycerol. The protein solution was applied to an IMAC metal chelate sorbent XK 26/20 column that was pre-equilibrated with 50 mM Tris, 0.5 M NaCl, 10% glycerol and 20 mM imidazole (pH 7.5) buffer. After sample application, the column was washed with 50 mM imidazole in a 50 mM Tris, 1 M NaCl and 10% glycerol (pH 7.5) buffer and the protein was eluted with 500 mM imidazole in a 50 mM Tris, 0.5 M NaCl and 10% glycerol (pH 7.5) buffer. Finally, the sortase A solution was desalted on a Sephadex G-25 XK 50/100 column with 50 mM Tris; 150 mM NaCl; 10% glycerin (pH 6.5) buffer.

#### 3.3.2. Enzymatic Immobilization of the BsrtA Protein onto a Matrix Using Sortase A

Samples of Amino-Sepharose 6FF (Cytiva) or modified Seplife 6FF matrices **7**, **10**, **11**, **14** and **15** (10 mL) were sequentially washed with 100 mL of different buffer compositions: 100 mM NaHCO_3_ (pH 8.43); 50 mM Tris (pH 8.0); 100 mM sodium acetate (pH 4.5); 50 mM Tris-HCl (pH 8.0). To the pre-washed matrix (10 mL), the following solutions were added step by step: a solution of BsrtA (9.5 mL of protein with a concentration of 21 mg/mL) in 50 mM Tris-HCl (pH 8.0); 1.25 mL of 4 M NaCl and the recombinant mutant sortase A (15.3 mL of protein with a concentration of 5.9 mg/mL). The protein concentration in the samples was determined spectrophotometrically. The total volume of the reaction mixture was brought to 50 mL with 50 mM Tris buffer (pH 8.0) and the mixture was stirred for 3 h at 37 °C. To calculate the molar ratio of proteins to resin in the reaction, 1 mL of the resin containing 17.5 µmol of amino groups was resuspended in 5 mL of a reaction mixture containing 1.28 µmol of BsrtA and 0.42 µmol of sortase A. Then, the sorbent was washed with the following buffers applying 100 mL of each buffer per 10 mL of resin: PBS (pH 7.4); 100 mM sodium acetate (pH 4.5); PBS (pH 7.4); 100 mM NaOH; 50 mM Tris-HCl (pH 8.0); and 20% aqueous solution of ethanol.

Determination of the amount of BsrtA protein ligated on the matrix was carried out using the Pierce BCA Protein Assay Kit (Thermo Fisher, Rockford, IL, USA). The measurements were performed on a Multiskan FC (Thermo Fisher Scientific, Shanghai, China) at a wavelength of 560 nm. Recombinant BsrtA protein was used as a standard for calibration.

### 3.4. Characterization of the Affinity Resins

#### 3.4.1. Estimation of the Dynamic Binding Capacity of the Resins (DBC_10_) Using Model IgG Antibodies

The estimation of DBC_10_ values was carried out using a column C10/10 (Pharmacia, Bromma, Sweden, size 10 cm × 10 mm). The volume of the applied resin sample was 2 mL.

The linear flow velocity corresponding to a given residence time (1, 3, 6 or 10 min) was calculated using the following formula:t = (H/υ) × 60(1)
where t is the residence time of the IgG with the resin in min; H is the height of the resin in the column in cm; υ is the linear flow velocity in cm/h.

The flow rate during the washing and elution steps was 2 mL/min. To determine the DBC_10_ values, a working solution of adalimumab A 2 mg/mL in PBS (pH 7.4) was applied. Measurement of the maximum absorption of an IgG protein solution with a known concentration was carried out using a spectrophotometer installed in the AKTA Pure instrument at a wavelength of 280 nm. After the affinity resin was equilibrated with 10 mL of PBS (pH 7.4), an aliquot of IgG was applied to the resin until an optical density equal to 10% of the optical density of the original solution was achieved. The dynamic capacity of the resin was calculated for the residence times of 1, 3, 6 or 10 min using the Dynamic Binding Capacity calculations software in UNICORN Extension. Finally, the resin was washed with 10 mL of PBS (pH 7.4) and bound IgG was eluted with 100 mM sodium acetate AT pH 3.5. After elution, the resin was washed with 10 mL of PBS.

#### 3.4.2. Cleaning-in-Place Procedure (CIP)

The resin’s (2 mL) resistance to cleaning was tested by its sequential washing with the following buffers at a rate of 0.5 mL/min: 15 mL of PBS (pH 7.4); 15 mL of 100 mM sodium acetate (pH 3.5); 15 mL of PBS; 15 mL of 100 mM NaOH; and 15 mL of PBS. Determination of the dynamic binding capacity (DBC_10_) was performed after every 20 cycles.

#### 3.4.3. Dependence of Pre-Column Pressure on the Flow Rate

The throughput of the sorbent was determined using a GE AKTA chromatographic system with an XK 16/40 column (GE Healthcare, Munich, Germany) with a diameter of 1.6 cm. A total of 30 mL of sorbent was packed into the column (column height about 15 cm). The change in the pressure was recorded when the linear flow rate in the chromatographic system was varied.

#### 3.4.4. Temperature Dependence of Dynamic Capacity

Estimation of DBC_10_ at different temperatures was carried out as described previously in the “Estimation of the dynamic binding capacity of the resins (DBC_10_) using model IgG antibodies” section. To maintain the assay temperature, a Lumex column thermostat (St Petersburg, Russia) was used.

## 4. Conclusions

In this work, we used sortase A as a biotechnological tool for the preparation of affinity resins for the isolation and purification of immunoglobulins G. Our synthetic protocol made it possible to ensure high density of the affinity ligand placement: about 6 mg (385 µmol) per 1 mL of modified agarose matrix. The resulting resin E by its main characteristics was comparable to the widely used commercially available resins. The dynamic binding capacity (DBC_10_) of model antibodies was up to 55 mg IgG per 1 mL of resin, whereby the resin retained its original capacity after 100 cycles of cleaning-in-place in the presence of 0.1 M NaOH. The obtained result indicates that the biocatalytic method for the immobilization of affinity ligands can be a good alternative to classical chemical methods of immobilization.

## Data Availability

The data presented in this study are contained within the article.

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
