# Peer review of "Affinity Resins for the Isolation of Immunoglobulins G Obtained Using Biocatalytic Technology"

_ijms, 2024, doi:10.3390/ijms25031367_

Round 1

Reviewer 1 Report

Comments and Suggestions for Authors

The article by Mikhail N. Tereshinet. al. entitled "Anity resins for isolation of immunoglobulins G obtained via biocatalytic technology " is an interesting study on the development of a biocatalytic method for production of novel affinity resins.

 The authors studied several conditions to obtain Bsrt A conjugates on agarose matrix. The conjugates were evaluated, and the results of main characteristics are similar to the widely used commercial resins.

In general, the article is written clearly and presents interesting data. This article would be of interest to scientists who are focusing on the study of biocatalytical technology.

I have the following comments and concern.

1.      Page 3, Table 1, Is there any explanation regarding difference between affinity resin A and B? What effects do you anticipate the positioning of EDC will have on DBC10?

2.      Page 8, line 313, What is the conjugation yield? In protein quantification, what protein was used for the calibration curve construction? What is the molar ratio of proteins to resin in the reaction? how did you decide that the reaction is completed?

3.      Page 9, line 376, What makes this biocatlytic method a good alternative? What are the advantages of this immobilization method compared to traditional chemical immobilization approaches?

Minor points

Page 3, Table 1, Please spell out the “EDA” and “BDDE”

“immunoglobulins G” would be IgGs.

Author Response

We should like to thank the reviewer 1 for critical reading of the ms and for the helpful advices, which have been considered for preparation of the revised version of the ms. The critical points raised by the reviewer were addressed on the point-to-point basis in this letter and the alterations are clearly labeled by yellow background in the main body text.

  1. Page 3, Table 1, Is there any explanation regarding difference between affinity resin A and B? What effects do you anticipate the positioning of EDC will have on DBC10?

We used GE Amino-Sepharose 6F to investigate the recognition capability of the sortase A enzyme towards primary amino groups (without the oligoglycin sequence). Having obtained a low DBC10 value, we decided to manually synthesize a matrix with various linkers. We utilized an agarose carrier from another company (Seplife 6FF) as a matrix, and ethylenediamine (EDA) as the first linker. According to the literature data (DOI: 10.1039/c7ra12128h), EDA is, in some cases, a more effective donor of the amino group in the transpeptidation reaction with sortase A. At the same time, we do not know the structure of the linker in the commercial sample GE Amino-Sepharose 6F. We hypothesize that the difference in DBC values is primarily attributed to the nature of the linker.

  1. Page 8, line 313, What is the conjugation yield?

The conjugation yield is approximately 30%, calculated based on values of immobilized BsrtA (about 6 mg per 1 ml of the resin) and initially added BsrtA to the reaction mixture (about 20 mg per 1 ml of the resin). L136

In protein quantification, what protein was used for the calibration curve construction?

Recombinant BsrtA protein was used as a standard for calibration. As recommended by the reviewer we add this sentence to the results section in L360.

What is the molar ratio of proteins to resin in the reaction?

Molar ratio of proteins to resin in the reaction: 1 ml of the resin containing 17.5 µmol of amino groups was resuspended in 5 ml of a reaction mixture containing 1.28 µmol of BsrtA and 0.42 µmol of sortase A. As recommended by the reviewer we add this information to the results section in L351-353.

How did you decide that the reaction is completed?

Having performed preliminary experiments, we established that the optimal reaction time is 3 hours. Experimental samples exhibited the highest DBC10 values at this duration. Shorter reaction time (1, 2 h) resulted in significantly lower DBC10, while longer duration (6;12; 24 h) did not lead to significant changes.

  1. Page 9, line 376, What makes this biocatlytic method a good alternative? What are the advantages of this immobilization method compared to traditional chemical immobilization approaches?

We follow the advice of the reviewer and included a new paragraph in which we argue in favor of proposed method (L171-196).

Reviewer 2 Report

Comments and Suggestions for Authors

L23: it seems that the resin was tested using only one antibody adalimumab, therefore the statement "assessed using recombinant hyperchimeric monoclonal antibodies" may be misleading.

abbreviation EDA should be explained

The discusion is unsatisfactory. The advatages of presented method should be discussed with literature. 

L303: bacterial cells (100 g) - it s quite uncommon to give the amount of cells in grams. Is it a weighted cell suspension?

Author Response

We should like to thank the reviewer 2 for critical reading of the ms and for the helpful advices, which have been considered for preparation of the revised version of the ms. The critical points raised by the reviewer were addressed on the point-to-point basis in this letter and the alterations are clearly labeled by yellow background in the main body text.

1)  L23: it seems that the resin was tested using only one antibody adalimumab, therefore the statement "assessed using recombinant hyperchimeric monoclonal antibodies" may be misleading.

We thank the reviewer for this comment, we corrected the sentence to the single case, L23.

2) abbreviation EDA should be explained

corrected as suggested by the reviewer, L100

3) The discussion is unsatisfactory. The advantages of presented method should be discussed with literature. 

We follow the advice of the reviewer and included a new paragraph in which we argue in favor of proposed method (L171-196).

4) L303: bacterial cells (100 g) - it’s quite uncommon to give the amount of cells in grams. Is it a weighted cell suspension

We are sorry for the misprint. We changed the term to “bacterial pellet”. L331.

Reviewer 3 Report

Comments and Suggestions for Authors

The manuscript " Affinity resins for isolation of immunoglobulins G obtained via biocatalytic technology" describes the development of a biocatalytic method for the production of novel affinity resins with immobilized mutant form of protein A via sortase A mediated reaction. It is an interested and well-developed study – the literature review (introduction) is a little bit short, but sufficient to give the readers an impression what the paper is about.

The material and method are properly described, reaction products are well characterized (I would, however, suggest including NMR spectra – in Supplementary Materials). Quantitative analysis of samples (e.g. determination of the number of oxirane groups attached to the matrix) is described sufficiently well (although some references could be added). Were any attempts made to run 13C or 2D NMR spectra?

Some minor issues that, in my opinion, should be improved are as follows:

1. Table 1 – it is a bit difficult to identify resins A to F from how they are described in the text – please list them in a more clear way

2. Scheme 1 is hard to read – please improve the quality, if possible.

In my opinion it would be an asset to explain the advantages of the methodology presented in this manuscript compared to the previously reported resins.

Author Response

We should like to thank the reviewer 3 for critical reading of the ms and for the helpful advices, which have been considered for preparation of the revised version of the ms. The critical points raised by the reviewer were addressed on the point-to-point basis in this letter and the alterations are clearly labeled by yellow background in the main body text.

The material and method are properly described, reaction products are well characterized (I would, however, suggest including NMR spectra – in Supplementary Materials). Quantitative analysis of samples (e.g. determination of the number of oxirane groups attached to the matrix) is described sufficiently well (although some references could be added). Were any attempts made to run 13C or 2D NMR spectra?

We follow the advice of the reviewer and additionally provide the 1H NMR spectra of Boc-Gly4-OH and Boc-Gly-Gly-EDA-Gly-Gly-Boc in the supplementary materials. We did not run 13C or 2D NMR spectra in this case since the number of protons of each group corresponded to the calculated number, and the obtained mass spectrometry data confirmed the molecular weight of compounds, but we will consider these methods in our other publications.

  1. Table 1 – it is a bit difficult to identify resins A to F from how they are described in the text – please list them in a more clear way.

We follow the advice of the reviewer and restructured the text in L123-126

  1. Scheme 1 is hard to read – please improve the quality, if possible.

We follow the advice of the reviewer. The Scheme 1 now is present as a high resolution image.

In my opinion it would be an asset to explain the advantages of the methodology presented in this manuscript compared to the previously reported resins.

We follow the advice of the reviewer and included a new paragraph in which we argue in favor of proposed method (L171-196).

Round 2

Reviewer 1 Report

Comments and Suggestions for Authors

Thank you for addressing the comments and concerns. I appreciate that my questions have been answered and properly added to the paper.